# Bilateral Maxillary Duplication in Tessier No. 7 Cleft: An Uncommon Congenital Deformity with a Challenging Radiological Diagnosis

**DOI:** 10.3390/diagnostics14070714

**Published:** 2024-03-28

**Authors:** Svetlana Antic, Djurdja Bracanovic, Aleksa Janovic, Goran Krstic, Djordje Plavsic, Biljana Markovic Vasiljkovic

**Affiliations:** Center for Radiological Diagnostics, School of Dental Medicine, University of Belgrade, 11000 Belgrade, Serbia; djurdja.bracanovic@stomf.bg.ac.rs (D.B.); aleksa.janovic@stomf.bg.ac.rs (A.J.); gorankrstic85@gmail.com (G.K.); drdjolep@yahoo.com (D.P.)

**Keywords:** maxillary duplication, Tessier 7 cleft, OPG, CT, congenital deformity

## Abstract

Tessier No. 7 cleft, known as lateral facial cleft, is a rare and understudied entity with an incidence of 1/80,000–1/300,000 live births. Besides perioral tissue abnormalities manifesting as macrostomia, Tessier 7 cleft also involves anomalies of the underlying bony structures. It can appear as part of a syndrome, such as Treacher-Collins syndrome or Goldenhar/Orbito-Auriculo-Vestibular Spectrum, or as an isolated form (unilateral or bilateral) with variable expressions. Bilateral maxillary duplication in Tessier 7 cleft is considered extremely rare, accounting for only two previously presented cases. Given that the cases presented in the literature mainly focus on clinical appearance and surgical treatment, without providing sufficient imaging, we aim to present key radiological features of Tessier 7 cleft in terms of evaluating the involved structures, which is essential for the therapeutic approach and final outcome. A 17-year-old male with incompetent lips and orthodontic abnormalities was referred to our Radiology Department for orthopantomography (OPG) and CT examinations. Hetero-anamnestic data revealed a history of surgical treatment of the commissural cleft conducted 2 months after the birth to enable feeding. Intraoral examination showed a maxillary cleft and supernumerary teeth. Since the given clinical presentation was inconclusive, radiological diagnostics took precedence in elucidating this complex entity.

**Figure 1 diagnostics-14-00714-f001:**
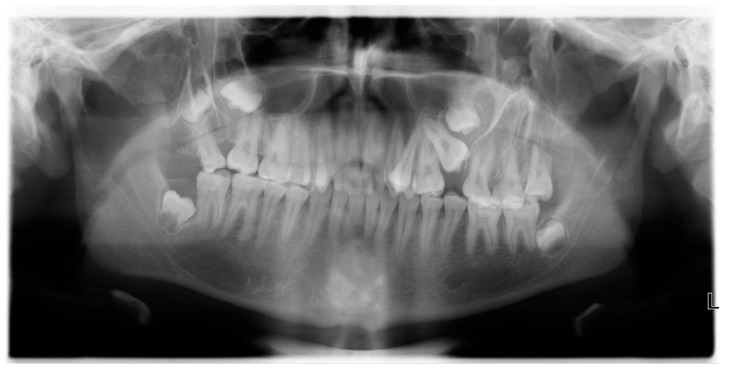
A 17-year-old white Caucasian male patient with incompetent lips as well as with jaw and teeth abnormalities was referred to our Radiology Department. The patient had no associated systemic diseases nor mental disorders. There was no reported family history of developmental disorders or inherited diseases either. Hetero-anamnestic data obtained from the patient’s parents revealed a history of both-sided lateral lip cleft (macrostomia) which was surgically treated 2 months after birth in order to enable feeding. Additionally, a few of the supernumerary teeth in the frontal maxillary region were extracted during the growth period. Very slight oblique facial linear depressions not crossing the anterior border of the masseter muscle were clinically observed, while an intraoral examination suggested a bilateral maxillary process cleft combined with supernumerary teeth. Orthopantomography (OPG), indicated by the orthodontist, revealed bilateral bony outgrowths located posterior to the maxillary tuberosity, thus simulating a posterior maxillary cleft. Each outgrowth, i.e., accessory maxillary process, contained supernumerary molar teeth—2 on the right and 3 on the left side. The patient was then referred to a maxillofacial surgeon, who indicated a computed tomography (CT) examination for a profound diagnostic evaluation.

**Figure 2 diagnostics-14-00714-f002:**
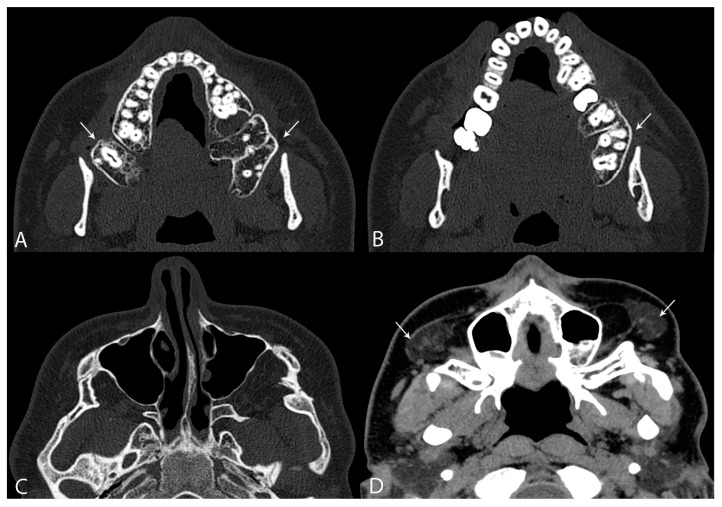
Given the limitations of 2-dimensional OPG and the suspected more complex deformity, including an anomaly of the zygomatic arch, an unenhanced CT examination was performed. An additional bilateral posterior maxillary process containing supernumerary teeth was confirmed (white arrows) (**A**,**B**). In addition, a bilateral abnormality of the zygomatic arch with incomplete dehiscence of the temporo-zygomatic suture was detected (**C**). Regarding soft tissues, skin ridging without muscular diastasis was noted, but bilateral accessory parotid gland tissue was present in the buccal region (white arrows) (**D**).

**Figure 3 diagnostics-14-00714-f003:**
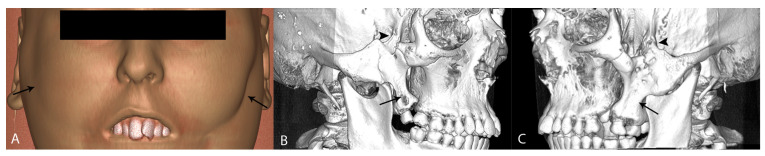
Soft tissue volume rendering revealed very slight oblique facial furrows (more prominent on the left side) not crossing the anterior border of the masseter muscle (black arrows) (**A**). A three-dimensional reconstruction of bony structures confirmed a duplicated posterior maxilla containing supernumerary teeth and a zygomatic bone deformity with additional outgrowths on the zygomatic arch. The zygomatic outgrowths were fused with the great wings of the sphenoid bone (black arrowheads), as well as with the additional maxillary processes (black arrows) (**B**,**C**). These clinical signs suggested a variant of Tessier No. 7 cleft, a rare entity with an incidence of 1/80,000–1/300,000 live births, or 0.3–1% cases within the cleft spectrum [1,2,3]. Resulting from abnormal fetal development of the 1st and 2nd branchial arches, it consequently leads to improper development of the perioral muscles of the face, thus appearing as macrostomia. Moreover, bony involvement is also present in the following structures: the pterygomaxillary and zygomatico-temporal junction, maxilla, greater sphenoid wings, or even mandibular condyle and coronoid [2,4,5]. It can appear as part of a syndrome, such as Treacher-Collins syndrome (TCS), Dandy–Walker syndrome (DWS), amniotic band syndrome (ABS), posterior fossa brain malformations, hemangioma, arterial lesions, cardiac abnormalities, and eye abnormalities (PHACE); and Goldenhar/Orbito-Auriculo-Vestibular Spectrum (OAVS). However, it can appear as an isolated form (unilateral or bilateral) with variable expressions [1,2,3,6]. The case presented here represents a rare case of an isolated bilateral cleft with maxillary duplication and no associated ear or mandibular abnormalities and muscle clefts, but with an associated parotid gland abnormality. The terms “maxillary duplication” and “accessory maxilla” refer to an extremely rare clinical entity that is characterized by the presence of an extra bone which is located posterior to the maxillary tuberosity and contains supernumerary teeth [4]. In the available literature, twenty-four patients with Tessier no. 7 clefts were described from 2000 to 2020, but only two of them presented with bilateral accessory maxillae, which was the case with our patient [1,3]. To the best of our knowledge, this is the first case report of Tessier No. 7 cleft from South Europe. Most of the cases presented in the literature were from developing countries, especially the Asian population [3]. Due to its low incidence, the variability of expression, and insufficient imaging support, we emphasize the typical radiological features indicating this rare entity.

## Data Availability

The data are contained within the main text of the manuscript.

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
