# Peer review of "Bilateral Maxillary Duplication in Tessier No. 7 Cleft: An Uncommon Congenital Deformity with a Challenging Radiological Diagnosis"

_diagnostics, 2024, doi:10.3390/diagnostics14070714_

Round 1

Reviewer 1 Report

Comments and Suggestions for Authors

hello

the following paper is a very interesting short paper focused on diagnostic images in Tessier cleft

its in the scope od diagnosis - short communication

tessier clefts are rare, troublesome in treatment, however all new images, radiographs and case descriptions are necessary in the literature to fully evaluate the spectrum of the disease and show other clinicians the value and meaning of such rare clefts

1 - title - good

the aspect of the title matches this short paper/radiographical study

all important aspects are presented in the title

2 - abstract - well structured, with brief introduction and case presentation

its aim - case report + radiological evaluation is clearly established

3 - key words -according to MESH

begining of the paper = good

4 - radiographs

are well prepared and are presenting a rare radiological features of tessier 7 cleft

both radiographs and ct are in good size and resolution

all anatomical and structural differences in this cleft are well presented

5 - figure1 

panoramic radiograph - good quality and well described

6 - figure 2

a structurised ct - a very clearly visible anatomical anomalies are presented

following figures are well designed

desciption is brief and adequate

7 - figure 3 3d rendering

following pictures are presenting rare features in 3d format

important elements are marked with an arrow

text is brief and well organised

8- further text

according to short communication/radiological study

meets the criteria to be brief and present most important features

9 - references are up to date 

please check style of paper presentation - minor changes, paper suitable for short communication

congratulations of an interesting case

with regards 

Author Response

Dear reviewer, thank you very much on your valuable comments.

Reviewer 2 Report

Comments and Suggestions for Authors

The manuscript addresses a modern and multidisciplinary research topic with an impact on the role of imaging investigation in diagnosis of Tessier no 7 cleft.  The authors sustain that the originality of the manuscript consists in radiological diagnosis presentation of an extremely rare case of an isolated bilateral Tessier no 7 cleft with maxillary duplication, excluding associated ear or mandible abnormalities and muscle tissue clefts, but with an associated parotid gland abnormality.

Author Response

(The authors gave the same response as above.)

Reviewer 3 Report

Comments and Suggestions for Authors

            Tessier number 7 facial cleft is a subject of great interest for specialists in the field of dentistry, oral-maxillo-facial surgery, reconstructive plastic surgery and not only. Unfortunately, in the presented article, the subject is not adequately addressed. The case could be better documented and presented in its evolution and the results of the surgical intervention could be compared with other similar cases.

            Discussion documentation is insufficient as is the bibliography. I recommend redoing the article.

Author Response

Dear reviewer,

this case is presented in the form of Interesting images, not classic Case report. Our patient, actually his parents (since he was 17 years old), unfortunatelly refused surgical treatment, fearing additional stress and possible complications of such a complex operation. Thus, we cannot compare the results of surgical intervention with other similar cases.

This article, designed in the form of interesting images, aim to present key radiological features, while previously presented cases mainly focus on the clinical appearance and surgical outcomes.

The bibliography contains two reviews, covering the majority of all the presented cases from the available literature.

Reviewer 4 Report

Comments and Suggestions for Authors

Comments on manuscript diagnostics-2912724, entitled “Bilateral maxillary duplication in Tessier No.7 cleft- an uncommon congenital deformity with challenging radiological diagnosis”.

Punctuation: change the hyphenation in the title after “cleft” to a coma.

The manuscript is required to be written in a scientific format, not just comments on radiographic images. The article should contain an introduction, case description, discussion, and conclusions. It should be written as a text and refer to Figures whenever required.

The description of the case needs to be mentioned in more detail such as demographic and systemic health.

Comments on the Quality of English Language

 Extensive editing of the English language required.

Author Response

Dear reviewer,

Thank you for the valuble comments, which significantly contributed to the improvement of our work.

We made changes according to your requests. We used trackchanges tool to mark all the changes that are made in the text.

1. We changed the hyphenation in the title after “cleft” to a coma.

2. This article is designed in the form of Interesting images, not classic case report. Thus it is adapted to the adequate format, according to the Diagnostics journal requirements. That is the reason why it does not contain sections such as: introduction, case description, discussion, and conclusions.

3. We added lines addressing patient‘s demographic data and systemic health, lines:32-35

4. We asked a colleague, a university professor of English for the extensive editing of the English language. We believe that the corrections we made have significantly contributed to the improvement of our work. Thank you.

Round 2

Reviewer 3 Report

Comments and Suggestions for Authors

I think it can be published.

Reviewer 4 Report

Comments and Suggestions for Authors

The author made the required correction. The article can be published in its current form.